# Ecological Flow Response Analysis to a Typical Strong Hydrological Alteration River in China

**DOI:** 10.3390/ijerph20032609

**Published:** 2023-01-31

**Authors:** Rui Xia, Hao Sun, Yan Chen, Qiang Wang, Xiaofei Chen, Qiang Hu, Jing Wang

**Affiliations:** 1State Key Laboratory of Environmental Criteria and Risk Assessment, Chinese Research Academy of Environmental Sciences, Beijing 100012, China; 2Laboratory of Aquatic Ecological Conservation and Restoration, Chinese Research Academy of Environmental Sciences, Beijing 100012, China; 3State Environmental Protection Key Laboratory of Estuarine and Coastal Research, Chinese Research Academy of Environmental Sciences, Beijing 100012, China; 4College of Urban and Environmental Sciences, Northwest University, Xi’an 710127, China; 5Chongqing Institute of Green and Intelligent Technology, Chinese Academy of Sciences, Chongqing 400714, China; 6University of Chinese Academy of Sciences, Beijing 100049, China

**Keywords:** hydrological alteration, ecological flow, DTVGM, flow duration gurve, Ganjiang River

## Abstract

Ecological flow is an important indicator for reflecting the stability of a watershed ecosystem. The calculation of ecological discharge under hydrological variation has become a research hot-spot. The Ganjiang River south of Poyang Lake in China was taken as an example in this study. Hydrological Alteration Diagnosis System methods were used to detect the change-points. The Distributed Time Variation Gain Model (DTVGM) was used to carry out runoff restoration. The Probability-weighted Flow Duration Curve was applied to calculate the ecological flow. The results showed that: (1) The hydrological alteration of the Waizhou Station occurred in 1991, the annual runoff increased by 10%, and the Gini coefficient (GI) increased by 0.07 after the change-point. The change in precipitation was the main driving factors. (2) The R value and NSE of the DTVGM were greater than 0.84, which represents the feasibility of the model used to restore runoff. (3) Compared to the traditional hydrological method, the proposed method can better reflect the inter-annual difference of ecological flow, flow ranges for high, normal, and low flow years are 398–3771 m^3^/s, 352–2160 m^3^/s, and 277–1657 m^3^/s, respectively. The calculation method of ecological flow in rivers considering hydrological variation can more scientifically reflect the impact of hydrological variation on ecological flow process, ecological flow under different human activities that can be calculated, such as dam control, water intake and water transfer, furthermore, it also provides a scientific basis for water resources planning and allocation under changing environment.

## 1. Introduction

Ecological flow is the key driver of river and lake ecosystem health, it generally refers to a flow process that maintains the health of the river ecosystem while also meeting human needs, and has played an important role in ensuring the spatial continuity and community diversity of the river ecosystem [1,2]. Meanwhile, with the loss of river connectivity and the drastic decline in river biodiversity, ecological flow has steadily drawn attention [3,4]. Climate change has increased the intensity and duration of regional extreme rainstorms and drought events, diminishing the value of river ecosystem services [5,6]. Human activities such as reservoir construction and urbanization have had a significant influence on the natural runoff process [7,8,9]. This massive river transformation has resulted in the channelization and discontinuity of rivers, as well as a considerable change in the natural evolution direction of rivers [10,11,12]. Changes in runoff consistency caused by climate change and human activities may result in unanticipated disasters in river ecosystems [13]. Owing to hydrological alteration, previous research on river ecological flow will be unable to be used for practical ecological management [14,15]. As a result, future eco-hydrology research should concentrate on the impacts of hydrological inconsistency on river ecosystems, particularly on aquatic animals.

According to data types and reference standards, the methods for calculating ecological flow are roughly classified into four categories: hydrological, hydraulics, habitat, and holistic methods [14,16,17]. Because the hydrological method completely relies on historical runoff data and does not need fieldworks, it is the most widely used, such as the northern great plains resources program (NGPRP), minimum monthly average flow (MMAF), monthly frequency computation (MFC), range of variability approach (RVA), and other methods. In recent years, some improved hydrological methods have also been proposed. For example, Tan et al., proposed a distribution flow method based on broadening kernel density estimation to calculate the ecological flow [18]. Long and Mei proposed a probability-weighted flow duration curve (PFDC) method to calculate the ecological flow for high flow, normal flow, and low flow years [19]. Wang et al., studied the ecological flow for ensuring the sustainable exploitation of mineral water in the Changbai Mountain basalt area using the Tennant method, the base flow ratio method, the driest monthly average flow method, and the Texas method [20]. However, the above studies did not consider whether the consistency conditions of hydrological series have changed. Meanwhile, while some scholars have conducted studies on ecological flow under hydrological alteration, most of the conclusions were based on the calculation of the runoff series prior to the hydrological alteration, which reduced the data utilization and the reliability of the results [21,22]. Furthermore, traditional hydrological methods only provide one or two recommended values of ecological flow, which is inappropriate because runoff varies greatly over different types of years and months, and the ecological flow must be adjusted accordingly. The calculation of ecological flow should take into account the intra-annual and inter-annual characteristics of runoff. Therefore, it is necessary to propose a new hydrological method to solve the problem of ecological flow calculation, especially in rivers with strong hydrological variation.

The Ganjiang River (GR) is the main tributary of Poyang Lake, which is the largest freshwater lake in China. The changes in the hydrological regime of GR have been very complicated under the influence of multiple relationships between rivers and lakes in past decades [23,24]. The Poyang Lake water surface usually shows “one slice” during dry season and “one line” during wet season under different water periods. Some studies have reported that hydrological alteration has occurred in the Poyang Lake basin, which has caused a significant impact on the survival of its aquatic organisms, particularly the fish of its indicator species [25,26,27,28]. Therefore, the goal of this study is to (1) diagnose the hydrological alteration of the GR’s runoff, (2) develop an ecological flow calculation method that can be applied to hydrological alteration conditions based on the extended natural runoff, and (3) analyze the causes of hydrological alteration and response characteristics of ecological flow. The study could provide a scientific foundation for local government reservoir ecological regulation, as well as a reference for calculating the ecological flow of other rivers under hydrological alteration.

## 2. Data and Methods

### 2.1. Study Area

The GR is located on the south bank of the Yangtze River’s lower reaches (113°34′–116°38′ E, 24°31′–8°45′ N), and is connected to the Yangtze River by Poyang Lake. The GR basin is defined in the study as the area above the Waizhou hydrological station, with a catchment area of 80,948 km^2^, accounting for 96.94% of the total catchment area. The Waizhou hydrological station, as a major hydrological control station of the basin, was selected for its hydrological variation and can be a mirror to reveal single or joint effects by climate change and accumulative influences of reservoirs on flow regimes. Located in a subtropical monsoon climate region, the GR basin, with area of 83,500 km^2^, has moderate temperatures all year and sufficient rainfall. The average annual temperature is around 18 °C, and the average annual rainfall is approximately 1580 mm. The basin’s inter-annual and intra-annual rainfall fluctuates significantly due to its location in the monsoon zone. Wet years have approximately double the rainfall of dry years, and the total rainfall during flood season is nearly half of the annual rainfall. For the purpose of fully exploiting hydropower resources and controlling floods, several reservoirs have been built along the mainstream, especially the Wan’an reservoir, which is the largest hydropower station on the GR. Figure 1 depicts the locations of the reservoir and the hydrological and precipitation gauges in the GR basin. 

### 2.2. Datasets

The daily rainfall data of 100 meteorological stations for 40 years (1979–2018) in the GR basin were provided by China Meteorological Assimilation Datasets for the SWAT model (CMADS) version 1.0 (www.cmads.org, (accessed on 1 June 2021)). CMADS V1.0 is a public dataset and its spatial resolution is 1/3° [29]. The daily meteorological products provided by CMADS have been validated for use in driving different hydrological models, including the Soil and Water Assessment Tool (SWAT), the Variable Infiltration Capacity model (VIC), the Storm Water Management model (SWMM), and others [30]. The Jiangxi Hydrological Bureau provided daily flow data from the Waizhou Hydrological Station for 65 years (1954–2018). A 3 arc-second (90 m) Digital Elevation Model (DEM) was collected from the Shuttle Radar Topography Mission (SRTM) Digital Elevation Database of USGS/NASA. (http://srtm.csi.cgiar.org/, (accessed on 1 June 2021)).

### 2.3. Methods

In this study, a new hydrological method called Restored Flow Duration Curve (RFDC) was proposed to study the ecological flow of rivers with strong hydrological variation. The RFDC method mainly combined the Hydrological Alteration Diagnosis System (HADS), the Distributed Time-Variation Gain Model (DTVGM), and the Probability-weighted flow duration curve (PFDC) method (Figure 2). The HADS was used to detect the change-point of hydrological time series with multiple hydrological alteration diagnosis methods. The DTVGM was introduced to restore the runoff after the hydrological alteration of the Waizhou Station, and the PFDC method was selected to calculate the ecological flow considering inter-annual and intra-annual flow changes.

#### 2.3.1. Hydrological Alteration Diagnosis Approach (HADS)

Hydrological alteration involves trend alteration and jumping alteration, so the hydrological alteration diagnosis can be divided into trend diagnosis and jumping diagnosis. In order to avoid the test error of single hydrological alteration detection methods and obtain reliable results, HADS, developed by Xie et al., was used to discern the change-point of the hydrological series [31]. HADS includes primary diagnosis, detailed diagnosis and comprehensive diagnosis. The primary diagnosis is based on Hurst coefficient and moving average method, and the detailed diagnosis includes trend diagnosis and jumping diagnosis. The comprehensive diagnosis is carried out based on the results of the preliminary and detailed diagnosis.

It should be noted that the occurrence of hydrological alteration should be analyzed in combination with the actual situation. Generally, the reason for hydrological alteration is mainly attributed to climate change and human activities. For the former, the change-point of the rainfall series can be identified to determine whether hydrological alteration is caused by meteorological alteration. The method is similar to the method used in hydrological alteration diagnosis. For the latter, it can be qualitatively judged by the Gini coefficient (GI) [32]. GI measures the degree of distribution uniformity of a set of data and can be used to assess the impact of water conservation projects on the distribution uniformity of runoff over the period of a year. The detailed calculation steps are given in Appendix A.

#### 2.3.2. Distributed Time-Variation Gain Model (DTVGM)

The DTVGM developed by Xia et al. [33] integrates the hydrological nonlinear theory and distributed hydrological model, which can simulate variant hydrological processes under different environmental conditions. It has been shown that DTVGM has shown promising results in more than 60 representative basins throughout the world, particularly in monsoon-prone areas [34]. The characteristics of DTVGM are as follows: (1) DEM grid and spatial digit information are used to characterize the time–space variation of precipitation, evaporation, air temperature, and land cover based on the DEM/GIS platform. (2) The soil moisture content links the two key processes of runoff generation and flow routing, allowing it to carry out a hydrological simulation based on grid elements and stream networks [35]. The core equation of DTVGM—the water balance equation—is shown as follows [36].
(1)$$∆AW_{t}=AW_{t+1}-AW_{t}=P_{t}-ETa_{t}-RS_{t}-RSS_{t}-WU_{t}$$
where ∆AW is the change of soil moisture storage, AW, in mm. The subscript t and t+1 represent variables at time step t and t+1, respectively. P is the precipitation, in mm. RS and RSS are the surface runoff and subsurface runoff, respectively. ETa is the real evapotranspiration. WU is the net water consumption, including water use, depression storage, ineffective evapotranspiration, and seepage loss. 

DTVGM has fewer parameters for runoff generation and the parameters can be estimated in terms of the system identification approach to reduce uncertainty [37]. Therefore, the Shuffled Complex Evolution (SCE-UA) algorithm was used to calibrate the selected sensitivity parameters [38], and two evaluation indexes, Nash–Sutcliffe efficiency coefficient (NSE) and correlation coefficient (*r*), were selected as evaluation indexes to evaluate the performance of the model. The higher the values of NSE and r near 1, the better the model performs.
(2)$$NSE=1-\frac{\sum_{i=1}^{n}{\left(Obs_{i}-Sim_{i}\right)}^{2}}{\sum_{i=1}^{n}{\left(\bar{Obs_{i}}-\bar{Sim}\right)}^{2}}$$
(3)$$r=\frac{\sum_{i=1}^{n}\left(Obs_{i}-\bar{Obs}\right)\left(Sim_{i}-\bar{Sim}\right)}{\sum_{i=1}^{n}{\left(Obs_{i}-\bar{Obs}\right)}^{2}\sum_{i=1}^{n}{\left(Sim_{i}-\bar{Sim}\right)}^{2}}\;$$
where Obs and Sim are observed and simulated values, respectively, and n is the number of values. $\bar{Obs}$ and $\bar{Sim}$ are averages of the observed and simulated values, respectively.

#### 2.3.3. Probability-Weighted Flow Duration Curve (PFDC)

The PFDC method provided by Long and Mei can calculate the ecological flow process under different types of years [19]. The main steps of the PFDC method are summarized as follows:

##### Frequency Analysis of Flow Series

By classifying the daily runoff series of no less than 20 years, the PFDC method obtains the annual runoff series and the monthly runoff series. To obtain the corresponding frequency distribution curve, the Pearson-III (P-III) distribution curve is employed here due to its wide application in China.

According to the frequency distribution curve, high flow years and months have a probability of average annual runoff (PN) and a probability of monthly runoff (PY) of less than 25%. Similarly, normal flow years and months are defined as PN and PY ranging from 25% to 75%, respectively; low flow years and months are defined as PN and PY more than 75%, respectively.

##### Calculation of the Initial Monthly Ecological Flow in Different Types of Years

The daily runoff data of different types of months are arranged in descending order to construct a daily flow duration curve. The monthly ecological flow is selected when the probability is 90%. Combining the ecological flow values under the same type for each month, the initial ecological flow for different types of years is obtained. 

##### Calculation of Probability Weights with Copula Functions

To construct the joint distribution of annual runoff probability and monthly runoff probability, five common two-dimensional Copula functions—Clayton, Frank, Gumbel, Gaussian, and Student t—are used. The joint distribution probability of various annual average runoff frequencies and monthly average runoff frequencies are then determined. By comparing the closeness of theoretical and empirical probabilities, and based on the principle of minimum root mean squared error (RMSE) and Akaike information criterion (AIC), an appropriate joint distribution function is selected.
(4)$$RMSE=\sqrt{\frac{\sum d_{i}^{2}}{n}}$$
(5)$$AIC=-2\ln\left(L\right)+2k$$
where n is the number of data; $d_{i}$ is the difference between empirical and theoretical probability; *L* is the likelihood function; *k* is the number of parameters.

The conditional probability formula is used to calculate the conditional probabilities for high flow months, normal flow months, and low flow months in different years. This conditional probability is then treated as the probability weight.

##### Calculation of the Final Monthly Ecological Flow in Different Types of Years

The final monthly ecological flow in high flow, normal flow, and low flow years are determined using the results of steps 2 and 3. 

## 3. Results

### 3.1. Hydrological Alteration Diagnosis

Based on multiple hydrological alteration diagnosis methods, the alteration trend of the annual flow series of the Waizhou Station from 1954 to 2018 was analyzed. The result of the moving average method is that the change-point of hydrological alteration is around 1990 while the Hurst coefficient is 0.7010 (Figure 3). Therefore, the preliminary diagnosis indicates that the hydrological series has a moderate alteration. By using the jumping diagnosis method and the trend diagnosis method of the HADS, we concluded that there is no trend alteration and the results of six jump diagnostic methods (i.e., Mann–Kendall method, Cumulative pitch average method, Pettitt method, Buishand U Test, Moving T test, and Mann–Whitney–Pitt method) showed that the break-point of hydrological alteration occurred in 1991. The detailed results are displayed in Table 1.

In fact, this conclusion was consistent with the research of Wang et al. [39]. According to the detailed diagnosis results, only the jumping diagnosis showed a significant trend among the trend and the jumping diagnosis. Therefore, the comprehensive diagnosis can be directly obtained without the need for comprehensive analysis of weights and significance. Overall, the hydrological alteration time of Waizhou Station can be determined to be 1991.

### 3.2. Runoff Restoration

Due to hydrological alteration, runoff series were divided into pre-altered and post-altered periods, and the distributed hydrological model was employed to restore the runoff during the post-altered period to obtain an extended natural runoff series. The pre-altered period from 1954 to 1990 was selected as the reference period to calibrate and verify the parameters of DTVGM. Taking into account the time scale of hydrological and meteorological data, the first eight years (1979–1986) were used as the calibration period, and the following four years (1987–1990) were used as the validation period. The SCE-UA algorithm is used to automatically optimize parameters and the key parameter values of DTVGM are summarized in Table 2.

Figure 4 depicts the simulation accuracy and runoff simulation results during the calibration and validation period. The NSE of the calibration and validation periods for the daily runoff simulation are 0.89 and 0.84, respectively, and the r are 0.95 and 0.92, respectively. Similarly, for the calibration and validation periods of the monthly runoff simulation, the NSE is 0.94 and 0.89, and the r is 0.97 and 0.95, respectively. The fitting curve shows a good consistency between the simulated and observed runoff, proving that DTVGM can simulate the GR’s natural runoff process. The daily runoff process from 1991 to 2018 was restored by fixing the DTVGM parameters, inputting the rainfall from 1988 to 2018, and setting the warm-up period from 1988 to 1990.

### 3.3. Ecological Flow Calculation

Based on the extended natural runoff series, the PFDC method was employed to calculate the ecological flow process in different types of years. After calculating the initial ecological flow process, the monthly conditional probabilities for high flow years, normal flow years, and low flow years are depicted in Figure 5. (Y, yearly; W-A, seasonal from winter throughout autumn; J-D, monthly from January throughout December. Winter corresponds the months from December to February). In high flow years, the probability of each month being a high flow month accounts for approximately 50%, and the sum of the probabilities of each month being a high flow month or a normal flow month account for approximately 90%. Similarly, in low flow years, the probability of each month being a low flow month is around 50%, and the sum of the probabilities of each month being a low flow month or a normal flow month is around 90%. In normal flow years, the probability of each month being a normal flow month is slightly higher than 50%, while being a low flow month or a high flow month is similar, respectively accounting for roughly 25%. In general, the GR’s runoff series show strong synchronicity of annual and monthly flow types, with the synchronicity tendency of these annual and monthly flow types being more prominent in spring and summer.

By treating the monthly conditional probabilities as the probability weight, the final ecological flow process in different types of years can be obtained (Figure 6). The value of ecological flow in the non-flood season (January to March and August to December) is reasonably close in different types of years, ranging from 300 to 1000 m^3^/s. On the contrary, the ranges of ecological flow during the flood season (April to July) are highly noticeable, and the maximum ecological flow in all types of years occurs in May. The ranges of ecological flow in the high flow years, normal flow years, and low flow years are 398–3771 m^3^/s, 352–2160 m^3^/s, and 277–1657 m^3^/s, respectively. It should be noted that the spawning period for the four major Chinese carps in the GR spans from April to July. Therefore, considering that the migration of fish and the growth of eggs depend on sufficient river flow, it is reasonable that the ecological flow during the flood season in different years shows such a large difference.

## 4. Discussions

### 4.1. Hydrological Alteration Analysis of GR

After applying various alteration diagnosis methods, it is shown that the annual runoff series of the GR has a break-point in 1991. As shown in Figure 3, the moving average line showed the phenomenon of “slow before and then steep”. Before 1991, the value of the moving average method fluctuated around the mean value of the runoff series, and then there was an obvious upward trend. This indicates that annual runoff has an increasing trend after hydrological alteration. The multi-year average flow (2269 m^3^/s) after the change-point increased by nearly 10% compared to the multi-year average flow (2075 m^3^/s) before the change-point. 

Climate change and human activities are the two most likely reasons of hydrological alteration. Based on relevant research findings, Lei et al., found that climate change is the primary factor of annual runoff changes in the Poyang Lake basin, where the GR is located, with climate change accounting for 91.88 % of the rise in runoff depth [40]. In addition, Guo et al., and Ye et al., came to similar conclusions [41,42]. Therefore, the hydrological alteration of GR is likely to be affected by the changing trend of the annual rainfall series. Table 1 shows the results of meteorological alteration diagnosis. It can be seen that the primary diagnosis shows that the annual rainfall series has a moderate alteration around 1991. Further analysis indicates that there is no trend alteration, but there is a jumping alteration in the annual rainfall series. The comprehensive diagnosis is that the annual rainfall series had meteorological variation in 1991. Thus, it can be concluded that meteorological alteration of the GR is the main reason for its hydrological alteration. 

Human activities have an impact on runoff by changing the earth’s surface and constructing water conservancy engineering. For the former, according to the research conclusions detailed above, the contribution of land use change to inter-annual runoff is small. Although land use change has no significant impact on the hydrological process of GR, for the hydrological process in some areas, the impact of conversion between different land use types cannot be underestimated. For the latter, considering the annual runoff change of the GR is very concentrated, the operation of water conservancy facilities will have an important impact on it. The construction of water conservancy facilities is not only used for flood control, power generation, and agricultural irrigation, but also to store part of the water during the flood season for ecological replenishment during the non-flood season, which will change the annual runoff distribution. The annual distribution uniformity of runoff can be characterized by the GI. The larger the GI, the more even the annual runoff distribution uniformity. Figure 7 shows the GI series and its moving average. Overall, the GI series shows a gradual increasing trend. The results of the moving average method show that the five-year moving line of the GI series increases sharply with time before 1991, and then gradually slows down after 1991. The mean values of the GI series were 0.59 and 0.66 before and after 1991, respectively. According to the information of water conservancy facilities, the Wan’an Reservoir, known as the GR’s first dam, was incorporated in August 1990, with a total storage capacity of 2.216 billion m^3^. Therefore, the operation of the Wan’an Reservoir may have had an impact on the runoff process in the coming year, especially during the dry season.

### 4.2. Ecological Flow Model under Strong Hydrological Alteration

Based on rainfall and runoff data, this study combined the three methods of HADS, DTVGM, and PFDC to calculate the ecological flow under hydrological alteration conditions. First of all, for hydrological alteration, the accurate diagnosis of the variation point is very necessary, which directly determines whether the subsequent steps are carried out and the correctness of the results. HADS integrates a variety of methods, and divides the hydrological alteration diagnosis into three steps: primary diagnosis, detailed diagnosis, and comprehensive diagnosis. It not only has high computational efficiency, but also has high accuracy in identifying the change-point of hydrological alteration. In addition, we analyzed the causes of hydrological changes based on the actual situation, which further verified the reliability of the results. Second, DTVGM was employed to extend natural runoff series. Although distributed hydrological models such as SWAT and MIKE SHE have been used by researchers to restore natural runoff after hydrological alteration, DTVGM has significant advantages compared to these models [39]. DTVGM does not require much data, such as land use data. Moreover, DTVGM requires less parameters to be calibrated and verified, and can automatically optimize parameters, thus reducing the difficulty of modeling. More importantly, the accuracy of its calculation can also meet the requirements. Finally, this study uses the PFDC method to calculate the monthly ecological flow in the high flow, normal flow, and low flow years. Compared to the traditional hydrology method, it can distinguish between inter-annual and intra-annual flow differences, thus the calculation results are more reasonable. A detailed discussion of the differences in ecological flow calculation results between the traditional hydrology method and the PFDC method can be found in Section 4.3.

There are two potential sources of uncertainty in our results: rainfall data and the scarcity of fish habitat data. For the former, the CMADS datasets used in this study started in 1979, while the hydrological alteration occurred in 1991. If more rainfall data can be collected, the accuracy of the runoff restoration will be further improved. For the later, due to the lack of habitat data of the four major Chinese carps in the basin, it is difficult to quantitatively evaluate the response of the flow changes to important habits such as fish reproduction and overwintering. Therefore, a method that can consider habitat information is required to verify the calculation results of the PFDC method in the future.

### 4.3. The Response Characteristics of Ecological Flow to a Strong Hydrological Alteration

In order to better evaluate the suitability of the ecological flow calculated by the PFDC method, traditional hydrological methods such as NGPRP, MMAF, MFC, and RVA were compared with the ecological flow calculated by the PFDC method (Figure 8). The ecological flow calculated by the MMAF method is the smallest, and its result is the closest to the PFDC method’s ecological flow in dry years, demonstrating the PFDC method’s reliability to a certain extent. The ecological flow calculated by the NGPRP, MFC, and RVA methods is particularly close to the PFDC method’s ecological flow in normal flow years. Overall, the calculation results of various methods are close in the non-flood season, ranging from 300 to 1000 m^3^/s. However, in the flood season, the calculation results of the traditional hydrological method are more cautious than the calculation results of the PFDC method. Due to the characteristics of the monsoon climate, the flood season flow of the Ganjiang River varies significantly under different types of years. For example, during the period we used to calibrate and verify the model parameters, the flood season flow is quite different (Figure 4b). This also shows that the ecological flow calculated by the traditional hydrological method is not always consistent with the real ecological water demand of the river in high flow and low flow years, and the calculation of ecological flow should consider the inter-annual variation of river flow for more accuracy.

The results of ecological flow calculation by PFDC method show that different recommended values of ecological flow should be provided according to the inter-annual and intra-annual characteristics of runoff in order to keep the GR ecosystem close to the natural variable flow process. Especially in the flood season, the inter-annual and intra-annual ecological flows differ significantly, which is related to the location of the GR in the subtropical monsoon region. For example, the ecological flow in May differs by approximately 1500 m^3^/s in high flow years and normal flow years. In addition, considering that meteorological alteration is the dominant factor increasing annual runoff of the GR, effective measures such as reservoir regulation should be taken to deal with the adverse effects of possible extreme meteorological events on the basin ecosystem. However, it should be noted that the reservoir projects cause the GR’s monthly runoff series to be flat, and the frequency of high flow is reduced correspondingly, which is adverse for the fish during the spawning period. As a result, it is critical to implement an effective ecological operation strategy to guarantee the ecological flow during the spawning period.

## 5. Conclusions

In this study, an integrated ecological flow method was developed to calculate river ecological flow processes under different flow types of years affected by hydrological alteration. The integrated method combines hydrological alteration diagnostic method, distributed hydrological model, and improved hydrological method. Our study is an important step forward in calculating ecological flow under hydrological alteration conditions. According to the comprehensive diagnosis results of HADS, the annual runoff series of the GR had hydrological alteration in 1991. In terms of the causes of hydrological alteration, meteorological alteration is the main factor, and human activities such as reservoir construction are secondary factors. The runoff series after the hydrological alteration is simulated by DTVGM, and the evaluation metrics indicate that the model has performance during both calibration and validation (NSE > 0.84, r > 0.92). The ecological flow calculated by the PFDC method is 398–3771 m^3^/s, 352–2160 m^3^/s, and 277–1657 m^3^/s in high flow, normal flow, and low flow years, respectively. Moreover, compared to the traditional hydrological methods, it can better reflect the intra-annual and inter-annual differences in ecological flow responses under different hydrological conditions, especially during the flood season (April–July). The insight gained from this study is that the increasingly frequent phenomenon of hydrological alteration should be taken into account when formulating river ecological flow. The method proposed in this paper provides technical support for the calculation of ecological flow under hydrological alteration conditions and other situations, such as human activities, dam construction, and intensive water use. The research results also provide a scientific basis for the planning and allocation of water resources in a changing environment.

## Figures and Tables

**Figure 1 ijerph-20-02609-f001:**
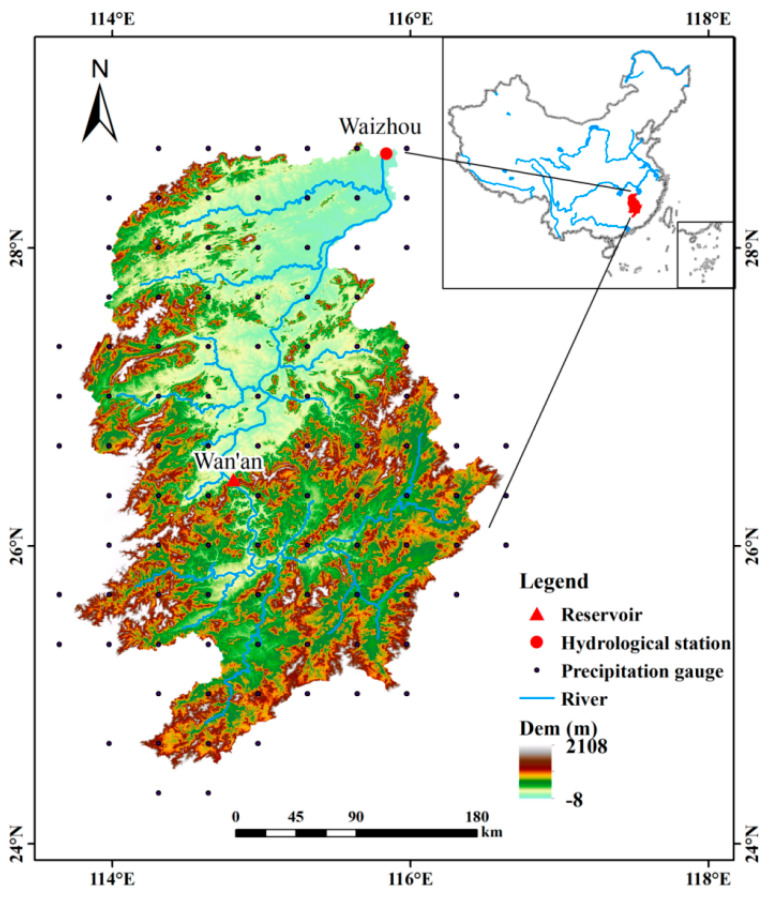
Location of the GR basin, reservoir, hydrological station, and precipitation gauges.

**Figure 2 ijerph-20-02609-f002:**
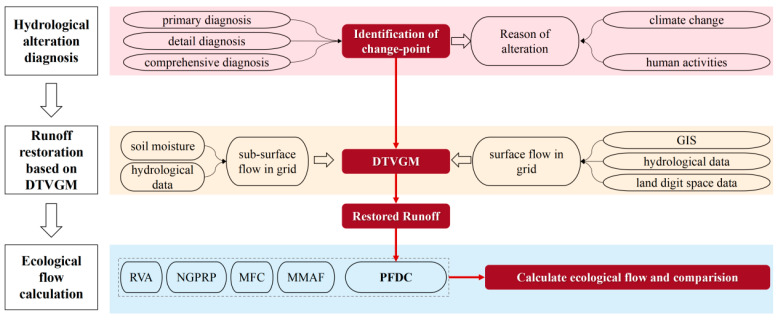
The ecological flow calculation framework based on RFDC.

**Figure 3 ijerph-20-02609-f003:**
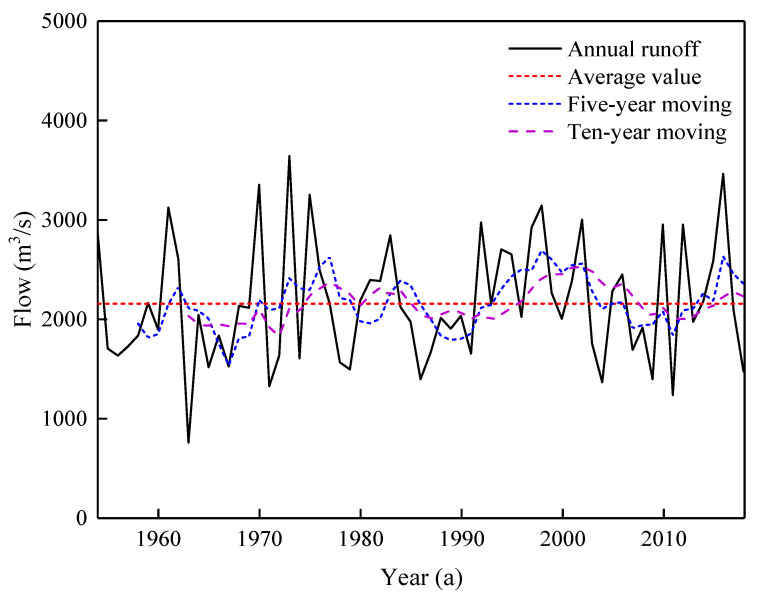
The series of annual average runoff and its moving series.

**Figure 4 ijerph-20-02609-f004:**
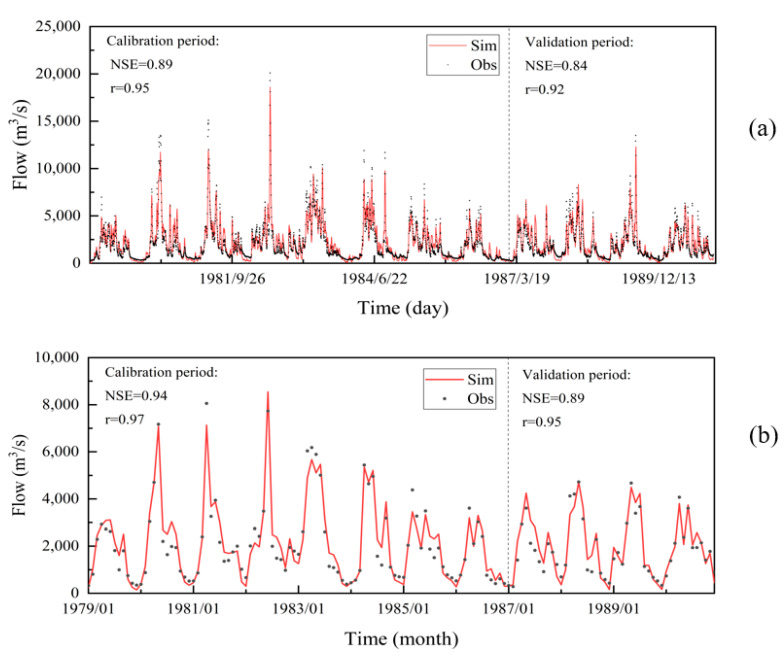
Simulation of (**a**) daily and (**b**) monthly runoff during the calibration and validation period in the GR basin (Sim: simulated values, Obs: observed values).

**Figure 5 ijerph-20-02609-f005:**
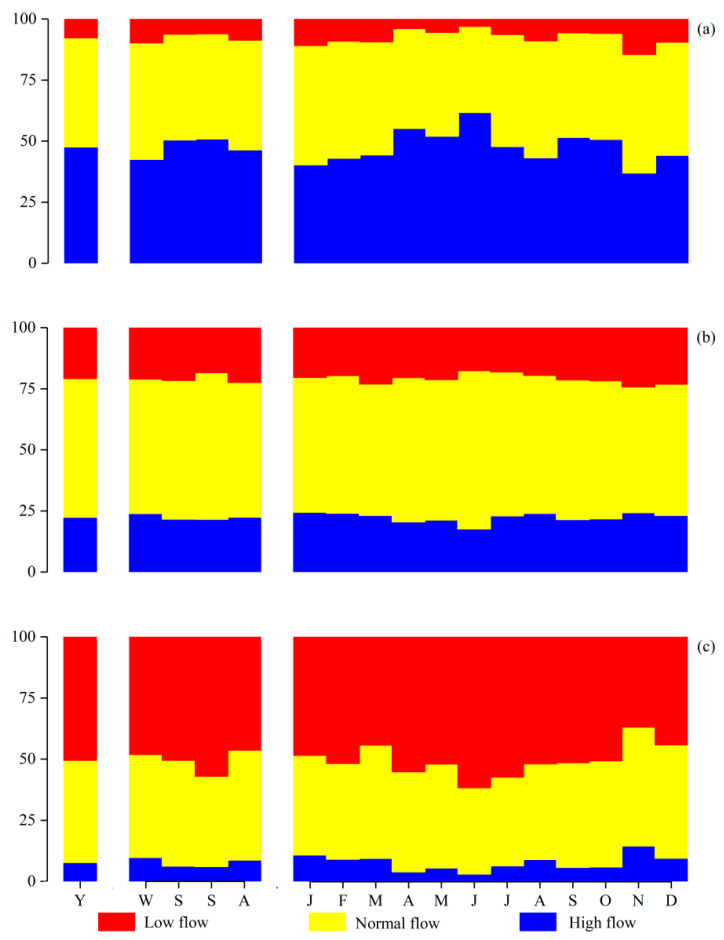
The monthly conditional probabilities in (**a**) high flow years, (**b**) normal flow years, and (**c**) low flow years.

**Figure 6 ijerph-20-02609-f006:**
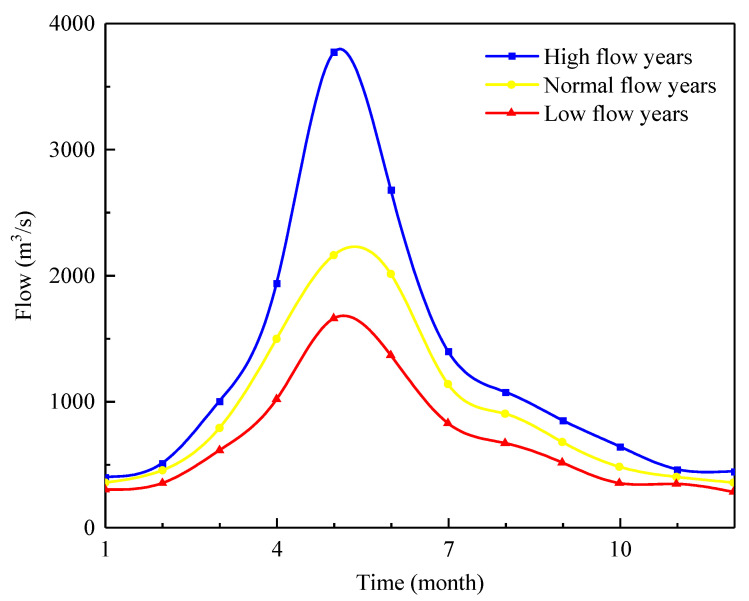
The ecological flow process in different types of years.

**Figure 7 ijerph-20-02609-f007:**
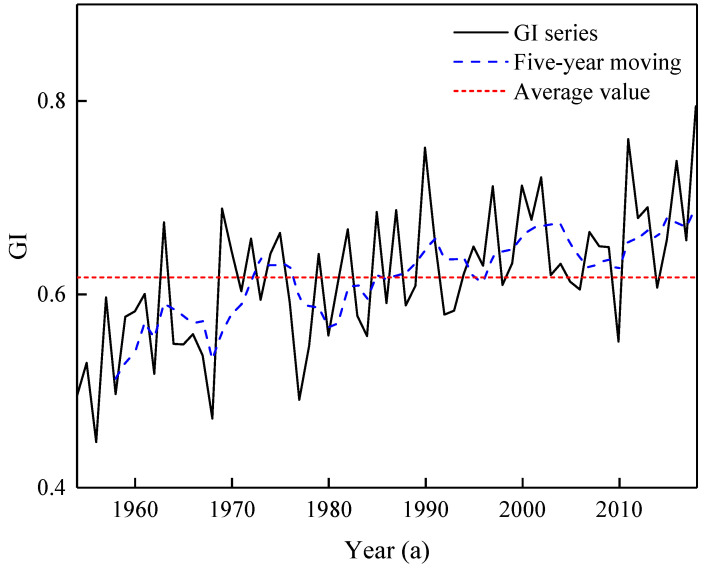
The GI series and its moving average.

**Figure 8 ijerph-20-02609-f008:**
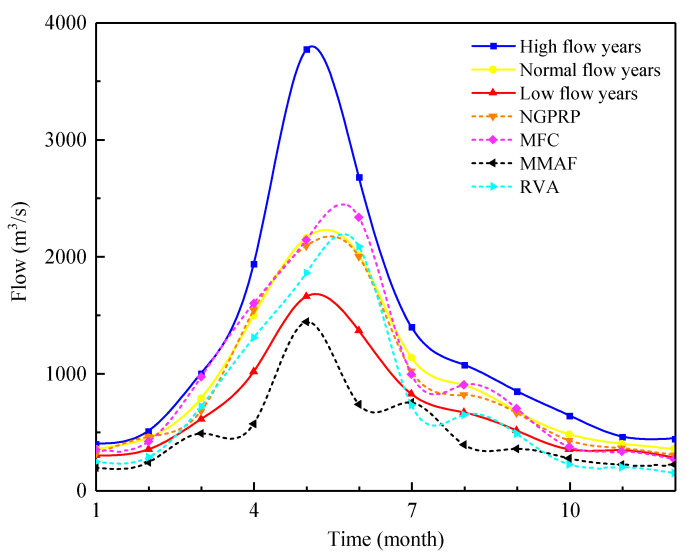
Comparison of ecological flows calculated by PFDC method and traditional hydrological methods.

**Table 1 ijerph-20-02609-t001:** Results of Hydrological and Meteorological Alteration Diagnosis.

Class		Diagnosis Method	Annual Runoff Series	Annual Rainfall Series
Primary diagnosis		Hurst coeffificient	0.7010	0.7074
		Moving average method	Around 1990	Around 1990
		Alteration degree	Moderate alteration	Moderate alteration
Detailed diagnosis	Trend diagnosis	Trend alteration degree	No trend alteration	No trend alteration
		Correlation coefficient method	No trend alteration	No trend alteration
		Spearman	No trend alteration	No trend alteration
		Kendall	No trend alteration	No trend alteration
	Jumping diagnosis	Cumulative pitch average method	1991	1991
		Mann–Kendall method	1991	1992
		Pettitt test	1992	1992
		Buishand U test	1991	1991
		Mann–Whitney–Pettitt test	1991	1991
		Lee–Heghinan method	1954	2014
		Moving T test	1991	1991
		Sliding F test	2016	2016
		Ordered cluster method	1967	1972
Comprehensive diagnosis			1991	1991

**Table 2 ijerph-20-02609-t002:** The key parameters in DTVGM.

Description	Value
Runoff parameter, g1	0.11
Runoff parameter, g2	0.05
Runoff parameter, g3	0.16
Surface confluence parameters, Nashn	2.92
Surface confluence parameters, Nashk	1.93
Linear reservoir confluence parameters, kkg	0.87

## Data Availability

Not applicable.

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
