# Peer review of "Ecological Flow Response Analysis to a Typical Strong Hydrological Alteration River in China"

_ijerph, 2023, doi:10.3390/ijerph20032609_

Round 1
Reviewer 1 Report
This paper focuses on the ecological flow methodologies and application in the Ganjiang River in the Poyang Lake watershed. It reads well and is very interesting. The hydrological simulation work provided valuable data for water resource management and water regulation. Some of the concerns that need to be addressed are:
1. In the abstract, the authors should provide more detail on how this method provides a scientific basis for water resource planning and allocation in changing environments.
2. The current abstract need to be improved, the authors should clearly show some background and more details of the results and their further implications for human-induced hydrological variation, eg water regulation.
3. In the conclusion, the authors concluded that this paper provides technical support for the calculation of ecological flow under hydrological alteration conditions, but I don’t think so, why don’t consider human activities, dam construction, and intensive water use?
Reviewer 2 Report
-I recommend revising the abstract to more concisely describes the key findings in the article
-In row 130 is written: "In this study, a new hydrological method called Restored Flow Duration Curve 130 (RFDC) was proposed". But in the following text this method is not mentioned at all, or does not elaborate further in the article. What´s the reason ?
Round 2
Reviewer 1 Report
The revised manuscript can be accepted.